# Detection of Protein Biomarkers Relevant to Sperm Characteristics and Fertility in Semen in Three Wild Felidae: The Flat-Headed Cat (*Prionailurus planiceps*), Fishing Cat (*Prionailurus viverrinus*), and Asiatic Golden Cat (*Catopuma temminckii*)

**DOI:** 10.3390/ani14071027

**Published:** 2024-03-28

**Authors:** Manita Wittayarat, Supalak Kiatsomboon, Navapol Kupthammasan, Wanlaya Tipkantha, Surasak Yimprasert, Ampika Thongphakdee, Saritvich Panyaboriban

**Affiliations:** 1Faculty of Veterinary Science, Prince of Songkla University, Songkhla 90110, Thailand; mwittayarat@gmail.com (M.W.); navapol.k@psu.ac.th (N.K.); 2Animal Conservation and Research Institute, The Zoological Park Organization of Thailand, Bangkok 10300, Thailand; supaluk.nb@hotmail.com (S.K.); wanlayav62@gmail.com (W.T.); 3Songkhla Zoo, The Zoological Park Organization of Thailand, Songkhla 90000, Thailand; ai_yares@outlook.co.th

**Keywords:** caspase-3, CRISP2, CRISP3, IZUMO1, semen, wild cat

## Abstract

**Simple Summary:**

In Thailand, the flat-headed cat, fishing cat, and Asiatic golden cat are among nine wild Felidae species requiring urgent conservation efforts. This study assessed routine sperm characteristics and detected protein biomarkers related to fertilization in these species. IZUMO1, involved in gamete interaction, showed the highest levels in the flat-headed cat. CRISP2 and CRISP3, crucial for sperm–egg fusion, had the highest levels in the fishing cat. Cleaved caspase-3, linked to DNA fragmentation, was found in all species and associated with decreased semen quality. These findings suggest that analyzing these biomarkers, alongside routine sperm characteristics, could enhance breeding management for wild Felidae, especially the flat-headed and fishing cats.

**Abstract:**

Effective wild cat conservation programs with assisted reproductive technologies are being developed in different parts of the world. The flat-headed cat, fishing cat, and Asiatic golden cat are three species among nine wild Felidae in Thailand that are in need of urgent conservation efforts. Here, we assessed routine sperm characteristics and we report the detection of protein biomarkers related to the fertilization process, IZUMO1 and the CRISP family, and apoptotic markers, active or cleaved caspase-3, in semen samples collected from these wild cats. IZUMO1 was located in the equatorial segment of the sperm head, which is the region involved in gamete interaction. The highest levels of IZUMO1 were found in both the sperm pellet and the seminal plasma of the flat-headed cat, as determined by immunoblotting. CRISP2, a sperm–egg fusion assisting protein, and CRISP3 were found in both the sperm pellet and the seminal plasma, and the highest levels were observed in the fishing cat. Positive correlations between certain semen parameters and IZUMO1, CRISP2, and CRISP3 expression were also demonstrated. Cleaved caspase-3 was found in all sperm samples in all three species and was associated with an increase in DNA fragmentation and a decrease in certain semen characteristics such as motility, viability, and intact acrosomes. Our results suggest that the analysis of IZUMO1, the CRISP family, and cleaved caspase-3, along with the routine sperm characteristics, may allow for better success in breeding management in wild Felidae, particularly in the flat-headed cat and the fishing cat.

## 1. Introduction

There are nine species of wild felids in Thailand that are classified as endangered, vulnerable, or near-threatened in the International Union for Conservation of Nature and Natural Resources Red List of Threatened Species. The flat-headed cats (*Prionailurus planiceps*) are one of the world’s least-known, with a distribution restricted to tropical lowland rainforests in peninsular Thailand, Malaysia, and the islands of Sumatra and Borneo, and whose future is highly threatened by the pollution of freshwater river systems and landscape fragmentation [1]. Fishing cats (*Prionailurus viverrinus*), meanwhile, are widely distributed among inland, wetland, and coastal areas of South and Southeast Asia, and they are currently classified as a vulnerable species on the Red List due to habitat loss and fragmentation [2]. Another examples is the Asiatic golden cats (*Catopuma temminckii*), which are found throughout the forests of mainland and insular tropical Asia and have been classified as near-threatened on the Red List due to forest loss and hunting [3]. Thus, there is currently an urgent need for conservation efforts, as these species are at risk of extinction.

Effective animal conservation strategies require fundamental knowledge of reproductive biology, genetics, and physiology. Investigation of the different functional proteins related to sperm characteristics and fertilization potential is important for anticipating the potential use of sperm for further reproductive purposes. A previous study has shown the co-evolution of several mammalian proteins such as IZUMO1 and the CRISP family, and these proteins are involved in sperm–egg fusion in the fertilization process [4]. The IZUMO1 protein in sperm is a member of the immunoglobulin superfamily that contains an extracellular immunoglobulin domain [4] and is a crucial mediator in the interaction and fusion of sperm with eggs. This protein was recently reported in cat spermatozoa and is detected at 17 kDa by immunoblotting [5]. However, IZUMO1 alone might not be sufficient to facilitate sperm–egg fusion; instead, it may need assistance from other proteins such as those of the CRISP family [6]. CRISP2, for instance, is a member of the cysteine-rich secretory proteins, antigen 5, and pathogenesis-related 1 protein (CAP) superfamily [4] and was reported to bind the fusogenic area of an egg [7,8]. Thus, we believe that the assessment of these proteins is essential for ensuring the ability of sperm to fertilize eggs, especially for endangered species where the number of females is limited. Although there is little information regarding the relevance of CRISP3 for fertilization [9], it is highly homologous to CRISP2 [8] and is mainly found in the seminal plasma. Furthermore, CRISP3 may have an important role in sperm protection during transit through the female reproductive tract, as shown in horses [10,11], suggesting that its evaluation may be beneficial. To our knowledge, however, no assessment of these proteins has been carried out in wild felids.

Male fertility is closely associated with not only sperm characteristics but also with apoptotic sperm biomarkers [12,13]. Certain caspases are central components of the apoptotic machinery, such as caspase-3, which is a member of the aspartic-acid-directed cysteine protease family [14]. Caspase-3 plays a pivotal role in the activation of death proteases and the catalyzation of the specific cleavage of many key cellular proteins [15]. Once caspase-3 is cleaved, the peptide end of this cleaved caspase represents a novel epitope called active or cleaved caspase-3, which is a major executioner of the apoptotic morphology [16]. Previous studies showed a positive correlation between cleaved caspase-3 in the sperm midpiece and DNA fragmentation of teratozoospermia and asthenozoospermia in human patients [17,18]; therefore, the results of a cleaved caspase-3 assay are another parameter to help determine sperm quality. In the present study, we aimed to elucidate the association between sperm characteristics and either an apoptotic sperm biomarker (cleaved caspase-3) or crucial proteins related to the fertilization process (IZUMO1 and the CRISP family) in three species of wild felids in Thailand: flat-headed cats, fishing cats, and Asiatic golden cats. The results may benefit further planning of conservation breeding.

## 2. Material and Methods

All the chemicals and reagents used in the present study were purchased from Sigma-Aldrich Chemicals Company (St. Louis, MO, USA) or Merck (Rahway, NJ, USA) unless otherwise indicated. All animal experimental protocols were in accordance with the requirements of the Animal Care and Use Protocol and were approved under Project NRIIS 4369803, issued by the Zoological Park Organization of Thailand under the Royal Patronage of H.M. the King. The animal procedure was carried out at the Songkhla Zoo (Songkhla province of Thailand, 7.14129° N, 100.60562° E).

### 2.1. Animals

Semen samples were obtained from three healthy, mature, captive flat-headed cats (*P. planiceps*) aged 12–14 years old and weighing 2.90–3.55 kg, three healthy, mature, captive fishing cats (*P. viverrinus*) aged 6–11 years old and weighing 9.70–13.85 kg, and one healthy, mature, captive Asiatic golden cat (*C. temminckii*) aged 11 years old and weighing 15.83 kg. All animals were housed individually in the zoo under the auspices of the Zoological Park Organization of Thailand (ZPOT) and were provided with high-quality preventive and clinical (veterinary) care.

Each animal received raw chicken meat supplemented with 1.7% (*w*/*w*) pre-mix and chicken eggs, fish, beef, pork, and rats, with an average daily feed intake of 200, 500, and 700 g/day in the flat-headed cats, fishing cats, and Asiatic golden cat, respectively. Vitamins and minerals were supplemented twice a week. Drinking water was always available. Information on the reproductive history, feeding, and other management practices of each animal was obtained from the animal managers.

### 2.2. Anesthesia and Morphometric Measurements

Male flat-headed cats, fishing cats, and Asiatic golden cats selected for semen collection via electro-ejaculation were restricted from feeding for 24 h and drinking for 12 h. All animals were anaesthetized intramuscularly by blow dart with the combination of ketamine HCl (7–10 mg/kg, Ketabel^®^, Bela-charm GmbH & Co., KG, Vechta, Germany) and xylazine HCl (0.5–1 mg/kg, L.B.S. Laboratory LTD., Bangkok, Thailand). Once the animals were recumbent, all animals were immediately intubated to facilitate positive-pressure ventilation, and the general anesthesia was maintained with 1–2% of isoflurane (Aerrane, Baxter Healthcare Corporation, Deerfield, IL, USA) for semen collection. Physiological parameters including heart rate, respiratory rate, pulse rate, body temperature, and blood oxygen saturation were monitored and recorded during anesthesia. After the procedure, all animals were given intramuscular injections of yohimbine HCl (0.125 mg/kg, Sigma-Aldrich Chemicals Company) to reverse the anesthesia and were closely observed until fully recovered.

### 2.3. Semen Collection and Evaluation

Semen collection was performed by using electro-ejaculation. A rectal probe (0.37 inches in diameter for flat-headed cat, 0.52 inches in diameter for fishing cat, and 0.77 inches in diameter for Asiatic golden cat) with longitudinal electrodes connected to an electrostimulator (Electro ejaculator e320, Minitube, Tiefenbach, Germany) was used. Briefly, the animal was placed in the lateral position and 80 electrical stimulations were applied while incrementally increasing the stimulation voltage (range, 2–5 V). Electrical stimulations consisted of three series: 2, 3, and 4 V (10 stimulations each), 3, 4, and 5 V (10 stimulations each), and 4 and 5 V (10 stimulations each), respectively. These were used as higher voltages such as 6 V have been challenged following cases in some animals, which were actively ejaculating. The stimulations were continued at the current voltage or gradually increased until ejaculation occurred. After semen collection, ejaculates from each individual animal with similar characteristics (sperm motility and viability) were pooled and were kept in a Styrofoam container stored at ambient temperature until the evaluation of sperm quality was performed. The ejaculate volume was estimated by pipetting, and semen pH was measured using pH indicator strips (EMD Millipore Corporation, Billerica, MA, USA). The sperm of domestic cats (*Felis catus*) was collected from the cauda epididymides of six castrated males aged 1–4 years old, provided by the Poonnakan Veterinary Care Clinic, Songkhla province, Thailand, with three cats exhibiting good sperm quality (motility ≥ 80%) and three cats exhibiting low sperm quality (motility < 40%). The sperm collection method followed the procedures outlined by Wittayarat et al. [5].

For each semen sample, the sperm concentration was quantified using an improved Neubauer hemocytometer (Superior, Bad Mergentheim, Germany). Two microliters of warm sperm suspension was placed on a pre-warmed 4X-CELL chamber slide (Cytonix, Beltsville, MD, USA; 20 µm depth) and at least 10 microscopic fields of each sample were immediately assessed for sperm motility using computer-assisted sperm analysis (CASA) on the HTR CEROS II Animal Motility version 1.11.9 (Hamilton Thorne Inc., Beverly, MA, USA).

Sperm viability was assessed by eosin–nigrosin staining, and at least 200 spermatozoa per male were evaluated. Intact membrane sperm was stained by eosin and appeared pinkish in color, whereas non-intact membrane sperm was not stained by the eosin stain and appeared white in color. We added 20 µL of semen to 500 µL of 4% paraformaldehyde for acrosomal membrane assessment using Coomassie Brilliant Blue staining. At least 200 spermatozoa per male were evaluated under bright field microscopy at 1000× [19,20]. Sperm with uniform blue staining overlying the acrosomal region were categorized as intact, sperm with patchy staining patterns over the acrosomal region were classified as damaged, and sperm that presented a clear, unstained area over the acrosomal region were defined as non-intact.

The plasma membrane function of the sperm was evaluated using the hypo-osmotic swelling test with 150 mOsm/kg of fructose and trisodium citrate dihydrate solutions. Assessments of at least 200 spermatozoa were made in three fields for each sample using a phase-contrast microscope. The plasma membrane integrity of the sperm was expressed as the percentage of sperm with curled tails (normal plasma membrane function) over the total number of spermatozoa. We fixed 20 µL of semen in 500 µL of 0.3% glutaraldehyde in phosphate-buffered saline buffer (PBS) for the assessment of sperm morphology using phase contrast microscopy at 1000× [21].

For the evaluation of DNA integrity, 5 µL of semen was spread on a glass slide, air-dried, and fixed in Carnoy’s solution (three volumes of methanol and one volume of acetic acid) overnight. The fixed slides were stained with acridine orange for 5 min, rinsed with water, covered with glass cover slips, and examined under a fluorescent microscope at 400× with the acridine orange (510–560 nm excitation) setting [22]. Spermatozoa with green fluorescence were considered as having a normal DNA content, whereas spermatozoa with yellow-orange to red fluorescence were considered as having damaged DNA.

### 2.4. Western Immunoblotting and Evaluation

Approximately 10% of the volume of each semen fraction was pooled to form a sample representing the whole ejaculate for further protein assessment. Semen samples were centrifuged at 800× *g* for 10 min to separate the sperm pellet and the seminal plasma, which were subsequently frozen at −80 °C until further analysis.

Sperm pellets were lysed in RIPA buffer (150 mM NaCl, 0.5% deoxycholate, 0.1% sodium dodecyl sulfate, 1% NP-40, and 150 mM Tris-HCl with a pH of 8.0) containing a protease inhibitor (5871S, Cell Signaling, Danvers, MA, USA). Protein concentrations of the sperm lysate and the seminal plasma were measured using UV absorbance (NanoDrop™, Thermo Fisher Scientific, Barrington, IL, USA) to adjust the amount of protein to 30 μg before being boiled for 10 min in Laemmli supplemented with β-mercaptoethanol. Equal total protein loads (30 µg) were loaded into each lane and were separated by 4–10% acrylamide gel electrophoresis at 100 V for 1.5 h, and then they were electrophoretically transblotted onto PVDF membranes (immobilon-P, Millipore) at 100 V for 1 h using the wet transfer method. A Pierce reversible-protein stain kit for PVDF membranes (Thermo Fisher Scientific, Rockford, IL, USA) was used according to the manufacturer’s instructions to detect the total protein, to normalize loading. The blots were incubated for 1 h at room temperature with Tris-buffered saline/0.1% Tween-20 (TBST) containing 10% (*w*/*v*) nonfat dry milk to block non-specific binding, and then either rabbit polyclonal anti-IZUMO1 (1:1000; Cat. No. ab211623, Abcam, Cambridge, UK), anti-CRISP2 (1:200; Cat. No. ab117442, Abcam), or anti-CRISP3 (1:10, Cat. No. ab105951, Abcam) antibodies were applied to the blots. The blots were then incubated with gentle shaking overnight at 4 °C. The negative control was performed using a matched immunoglobulin G (IgG) isotype control for rabbits (Cat. No. ab171870, Abcam). After washing, the blots were subsequently probed with secondary alkaline phosphatase-conjugated goat anti-rabbit antibody (1:4000; Cat. No. ab97051, Abcam) for 1 h at room temperature, followed by immunodetection with ECL reagent (Cat. No.170-5061, Bio-Rad Laboratories, Richmond, CA, USA). The protein bands were visualized using enhanced chemiluminescence (ECL; Thermo Fisher Scientific, Waltham, MA, USA) and developed by an Image Quant LAS 500 (GE Healthcare Biosciences AB, Uppsala, Sweden). Total and target protein band intensities were quantified using Image Quant TL software (version 8.2.0.0, GE Healthcare Biosciences AB, Uppsala, Sweden). Total protein was normalized using a method modified from our previous work [5,23].

### 2.5. Sperm Immunofluorescence Microscopy

To assess the protein localization patterns of IZUMO1, CRISP2, and CRISP3 in wild cat spermatozoa, sperm samples were fixed in 3.7% paraformaldehyde in PBS overnight at 4 °C, permeabilized with 0.1% (*v*/*v*) Triton X-100/PBS for 40 min, and stored in 1% (*w*/*v*) BSA/PBS. The permeabilized sperm were blocked against non-specific binding with 10% goat serum/PBS for 1 h before incubation with primary rabbit polyclonal anti-IZUMO1 (1:300), anti-CRISP2 (1:200), or anti-CRISP3 (1:10) antibodies in a moisture chamber overnight at 4 °C, followed by incubation with Alexa 488-conjugated goat anti-rabbit IgG secondary antibody (1:500; Cat. No. ab150077, Abcam) for 1 h. The samples were then mounted with ProLong Gold Antifade reagent and DAPI (molecular probes, Life Technologies, Eugene, OR USA) on glass slides for nuclear DNA staining. A matched IgG isotype control for rabbit was used as the negative control. At least 100 spermatozoa of each sample were imaged for protein localization using a fluorescence microscope (Nikon Eclipse Ci-L; Nikon, Tokyo, Japan) at 1000× under the Alexa Fluor® 488 (330–380 nm excitation) and DAPI (450–490 nm excitation) settings. Images were subsequently acquired with a Nikon Digital Sight DS-QiMc (Nikon, Tokyo, Japan) equipped with the NIS-Elements BR Ver 4.20.02 imaging software package running on a workstation.

To assess the apoptosis rate of sperm cells, permeabilized sperm were stained at 4 °C overnight with a primary rabbit polyclonal anti-cleaved caspase-3 antibody (1:100; Cat. No. ab2302, Abcam), followed by an Alexa 488-conjugated goat anti-rabbit IgG secondary antibody (1:500). The samples were then mounted with ProLong Gold Antifade reagent and DAPI on glass slides. Samples were observed under a fluorescence microscope with Alexa Fluor^®^ 488 (Thermo Fisher Scientific, Waltham, MA USA) and DAPI settings and counted to quantify the apoptosis rate. Sperm stained with Alexa Fluor^®^ 488 were classified as immunopositive sperm. The percentage of immunopositive sperm was determined by counting 200 spermatozoa twice throughout the slide under a high-power field at 1000×. Sperm cells from domestic cats with good sperm quality (motility ≥ 80%) and low sperm quality (sperm motility < 40%) were used for comparison.

### 2.6. Statistical Analyses

All statistical analyses were performed using the R programming language (version 4.1.2) and R studio (version 2022.07.0+548). The data are presented as mean ± standard error of the mean (SEM) unless specified otherwise. The normality of the data was assessed using the Shapiro–Wilk test. To compare semen characteristics, dependent variables were analyzed using the Wilcoxon rank-sum test. Correlation analysis (Pearson’s for normal distribution or Spearman’s for non-normal distribution) was employed to assess the relationship between protein expression in spermatozoa and seminal plasma with semen characteristics. All *p*-values lower than 0.05 were considered statistically significant.

## 3. Results

### 3.1. Differences in Ejaculated Semen Characteristics and Sperm Traits among Three Wild Cat Species

The semen collection procedure was successful for every male across all species, but there was a wide variation in semen characteristics among the species and individual males. The overall ejaculated semen characteristics of the three felid species including the flat-headed cat (three males), fishing cat (three males), and Asiatic golden cat (one male) are shown in Figure 1 and Figure 2.

Both the mean ejaculate volume and sperm concentration were highest in the fishing cats (250.2 µL and 32.6 × 10^6^ spermatozoa/mL) when compared to those of the flat-headed cats (147.2 µL and 20.5 × 10^6^ spermatozoa/mL) and Asiatic golden cat (136.0 µL and 0.1 × 10^6^ spermatozoa/mL); however, significant differences in mean ejaculate volume and sperm concentration were only observed between the fishing cat and Asiatic golden cat (*p* < 0.05). The highest abnormality of sperm morphology (67%), which was classified into primary (15%) and secondary abnormal defects (43%), was detected in the Asiatic golden cat compared to the other species (*p* < 0.05). The most common sperm abnormalities of all three wild cat species in this study were a bent midpiece, a bent tail, and a tightly coiled tail, which are shown in Figure 3.

### 3.2. Expression of IZUMO1, CRISP2, and CRISP3 Proteins in Spermatozoa and Seminal Plasma among Three Wild Cat Species

In Western immunoblotting, IZUMO1, CRISP2, and CRISP3 were detectable in both the spermatozoa and the seminal plasma of all wild cat species; however, the expression levels differed among the species. The apparent molecular sizes of IZUMO1, CRISP2, and CRISP3 detected in all three species were approximately 17, 25, and 29 kDa, respectively. The expression levels of IZUMO1, CRISP2, and CRISP3 in both the sperm pellet and the seminal plasma in each species are shown in Figure 4.

There was a significantly higher ratio of IZUMO1 to total protein in both the sperm pellet and the seminal plasma in the flat-headed cat, followed by the fishing cat and then the Asiatic golden cat (*p* < 0.05). The level of IZUMO1 in flat-headed cat spermatozoa was found to be lower than that in domestic cats with good sperm quality (6.80 ± 0.65) (*p* < 0.01), but it was found to be higher than that in domestic cats with low sperm quality (1.14 ± 0.19) (*p* < 0.05). When spermatozoa from the three wild cats were immunostained with an antibody against IZUMO1, the fluorescent luminous sites of spermatozoa were predominantly detected at the equatorial segment of all species (Figure 5), which was inconsistent with our previous study in domestic cats [5].

For CRISPs, both CRISP2 and CRISP3 had the highest expression levels in both the sperm pellet and the seminal plasma in the fishing cat, followed by the Asiatic golden cat and then the flat-headed cat (*p* < 0.05). Similar to the results for IZUMO1, the protein expression levels of CRISP2 and CRISP3 in fishing cats fell within the intermediate range between good sperm quality (7.11 ± 0.10 and 3.64 ± 0.06, respectively) and low sperm quality (2.60 ± 0.05 and 1.33 ± 0.02, respectively) in domestic cats, with statistically significant differences (*p* < 0.05). The immunostaining signals of CRISP2 and CRISP3 were mainly expressed in the sperm head (Figure 6).

### 3.3. Correlation between IZUMO1, CRISP2, and CRISP3 Protein Expression Levels and Ejaculate Characteristics among Three Wildcat Species

The correlation of each protein expression level (IZUMO1, CRISP2, or CRISP3) in both the sperm pellet and the seminal plasma with ejaculate characteristics is shown in Table 1. Positive correlations were observed in either the sperm pellet or the seminal plasma between IZUMO1 and all ejaculate semen parameters except for semen volume. CRISP2 expression was positively correlated with the percentages of sperm motility, viability, membrane integrity, and normal DNA integrity. On the other hand, CRISP3 levels were positively associated with semen volume, concentration, total sperm number, and sperm membrane integrity.

### 3.4. Percentage of Cleaved Caspase-3-Positive Sperm and Its Correlation with Semen Characteristics among Three Wildcat Species

Figure 7 displays positive immunofluorescence images of cleaved caspase-3 in flat-headed cats, fishing cats, and Asiatic golden cats. The fishing cat (27.7 ± 1.8%) displayed a lower percentage of positive cleaved caspase-3 in spermatozoa compared to the flat-headed cat (43.7 ± 3.9%) and the Asiatic golden cat (52.3 ± 0.6%). When comparing these results to domestic cats, both the fishing and flat-headed cats displayed a higher percentage of positive cleaved caspase-3 than domestic cats with good sperm quality (12.67 ± 2.41%) (*p* < 0.05), but there was no significant difference when compared to domestic cats with low sperm quality (48.67 ± 3.82%). Meanwhile, the Asiatic golden cat exhibited the highest percentage of positive cleaved caspase-3 compared to both good and low semen qualities in domestic cats (*p* < 0.01).

The expression of cleaved caspase-3 in spermatozoa was negatively correlated with the percentage of sperm motility (r = −0.417, *p* < 0.01), sperm viability (r = −0.531, *p* < 0.01), intact acrosomes (r = −0.717, *p* < 0.01), and normal DNA (r = −0.604, *p* < 0.01); however, there was no correlation with semen volume, concentration, number of total spermatozoa, or morphology (*p* > 0.05).

## 4. Discussion

In the present study, we were able to detect proteins related to sperm characteristics and fertilization potential, including IZUMO1 and the CRISP family (CRISP 2 and 3), in the sperm and seminal plasma of all three wild cat species investigated. Our previous work was the first to successfully identify IZUMO1 in domestic cat sperm at a molecular weight of approximately 17 kDa, and the same size was detected in wildcats in this study; however, the status of this protein in seminal plasma was not previously assessed since the sperm samples were obtained from cat epididymis [5]. Based on the results of other work in ram, IZUMO1 was found in the seminal plasma via the analysis of quantitative proteomics data [24]; thus, we predicted that IZUMO1 may also be present in the seminal plasma of wild cat species.

To test this prediction, the expression of IZUMO1 was examined in the seminal plasma collected from three wild cat species and the results showed the presence of IZUMO1 in all seminal plasma samples tested. It has been reported that in horses, several proteins in the seminal plasma including clusterin, CRISP3, epididymal sperm-binding protein 1, kallikrein1E2, seminal plasma protein A3, and HSP interact with the sperm membrane and are related to binding and catalytic activities [25]. Although IZUMO1 in the seminal plasma has not yet been reported to interact with the sperm membrane, it is tempting to speculate that this protein could be involved in a series of biochemical and structural changes implicated in sperm function and gamete interaction, since it was expressed in all samples. Therefore, further protein functional analysis of IZUMO1 in the seminal plasma is required to gain insights into the role played in male reproductive physiology.

The highest level of IZUMO1 in sperm was found in the flat-headed cat, which was in between that of the domestic cat with good sperm quality and that of the domestic cat with poor sperm quality, indicating that IZUMO1 levels in the sperm were similar between the flat-headed cat and the domestic cat. Immunofluorescent staining revealed that IZUMO1 was predominantly localized at the equatorial segment of the sperm head, which is the region considered to be important in gamete interaction as fusion between the spermatozoa and the oocyte during fertilization occurs at this site [26]. Moreover, all semen parameters except for semen volume were positively correlated with IZUMO1 levels in either the sperm pellet or the seminal plasma. These data suggest that the detection of IZUMO1 both in the spermatozoa and the seminal plasma may enable the prediction of fertility potential in felid populations, especially in the flat-headed cat.

CRISPs are relevant for sperm physiology since they are mainly expressed in the male reproductive tract in humans [27], rodents [27], horses [28], and Malayan tapirs [23]. An aberrant expression of CRISP2 is associated with male infertility due to its participation in gamete fusion [27]; however, to our knowledge, there are no studies on CRISP (CRISP2 and CRISP3) expression in the semen of felid-related species. We showed by immunoblotting that CRISP2 and CRISP3 can be found in both the sperm pellet and the seminal plasma, and the highest level was observed in the fishing cat. Fishing cats may have similar levels of CRISP2 and CRISP3 to the domestic cat, which we surmise because expression levels in the fishing cats were in between the good- and poor-sperm-quality cats. Interestingly, it seems that the species with low levels of IZUMO1 were likely to have high CRISP2 levels instead, potentially to assist in the sperm–egg fusion process. This may be supported by the previous report that IZUMO1 and other proteins like CRISP2 are necessary to enable sperm–egg fusion [6]. CRISP2 is stored in the acrosome in humans and rodents and is released when the sperm encounters the zona pellucida [29]. We believe that this protein is also likely to be found at the acrosome region of felid species, since CRISP2 protein expression tended to have a positive correlation with the percentage of intact acrosome sperm (*p* = 0.089) and the data showed a localization of CRISP2 at the sperm head.

CRISP3 is a major protein found in the seminal plasma [29]; therefore, the intense CRISP3 immunoreactivity observed in the sperm pellets from the fishing cat and the Asiatic golden cat was unexpected. Since CRISP3 is closely related to CRISP2 in terms of high amino acid sequence identity (72% identity in humans) [29], some of the immunoreactivity observed in the sperm pellets is possibly due to cross-reactivity with CRISP2. In this study, however, all sperm samples were found to produce consistent and clearly distinguishable bands between CRISP2 and CRISP3, as the detected molecular weight of CRISP3 was around 29 kDa, which was approximately 4 kDa higher than that of CRISP2 (25 kDa). This afforded the same results as our previous report in Malayan tapirs [23]. Moreover, previous studies have also reported that human CRISP2 exists in only one molecular weight form [30,31], and that human CRISP3 can be found in mature spermatids and spermatozoa [29]. Taken together, these findings suggest that CRISP3 signals found in the sperm pellets were not from cross-reactivity with CRISP2. The function of sperm CRISP3 is not well-understood, but is possibly relevant to the sperm fusion ability in humans [9]; however, further studies are still needed. CRISP3 in the seminal plasma was found to be the only protein in this study that was positively correlated with semen volume, and we propose that it probably helps to protect sperm during transit in the female reproductive tract. Thus, a similar finding is likely in horses [10,11]. The analysis of CRISP2 and CRISP3 in semen, along with other sperm characteristics, will, therefore, be useful for wild felid conservation planning, at least in the fishing cat, where their population in the zoo is still viable.

Semen samples with increased rates of sperm DNA fragmentation are related to decreased rates of fertilization [32] and pregnancy [33]. Previous studies have shown a positive correlation between cleaved caspase-3 and DNA fragmentation [17,18]. Immunofluorescent staining mainly showed cleaved caspase-3 in the sperm head and the midpiece of the sperm tail, which is in an agreement with the study by Mohammadi et al. [34]. Although a few previous studies have reported observing cleaved capase-3 localization exclusively in the sperm midpiece region, our data revealed that the percentage of cleaved caspase-3 was positively correlated with DNA fragmentation, indicating the apoptosis can occur simultaneously in the sperm head where the chromatin and DNA are stored [34]. Moreover, a significant negative correlation between cleaved caspase-3 immunostaining and sperm motility was demonstrated in this study. Similar to our results, caspase-3 activity has been shown to be significantly associated with low sperm motility, concentration, and morphology in ejaculated human sperm [17,35]. These results suggest that in ejaculated sperm from wild felids, cleaved caspase-3 is present and may function to increase DNA fragmentation, decrease semen quality, and consequently, cause low rates of fertilization and pregnancy. However, it is worth noting that limitations of this study include a small sample size and variations in the age of the animals, which could influence the generalizability and interpretation of some reproductive parameters and protein expression levels. Future studies with larger, age-matched groups are needed for validation.

## 5. Conclusions

In conclusion, the analysis of crucial proteins related to the fertilization process, IZUMO1 and the CRISP family, and the apoptotic sperm biomarker cleaved caspase-3 may support a more accurate prediction of breeding success than the use of routine sperm characteristics alone. This information may help wildlife conservation veterinarians to plan breeding programs, especially in species where the number of females is limited.

## Figures and Tables

**Figure 1 animals-14-01027-f001:**
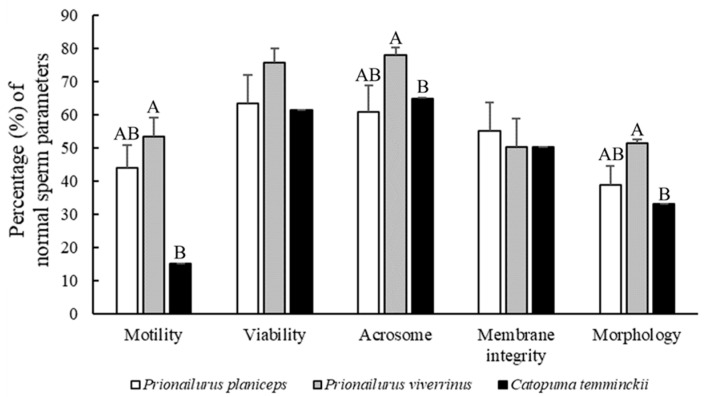
Comparisons of ejaculated semen traits among the flat-headed cat (*Prionailurus planiceps*), fishing cat (*Prionailurus viverrinus*), and Asiatic golden cat (*Catopuma temminckii*) in various parameters, including percentages of sperm motility, viability, acrosomal integrity, membrane integrity, and normal morphology. Bars with different superscripts within the same row are significantly different across species (*p* < 0.05).

**Figure 2 animals-14-01027-f002:**
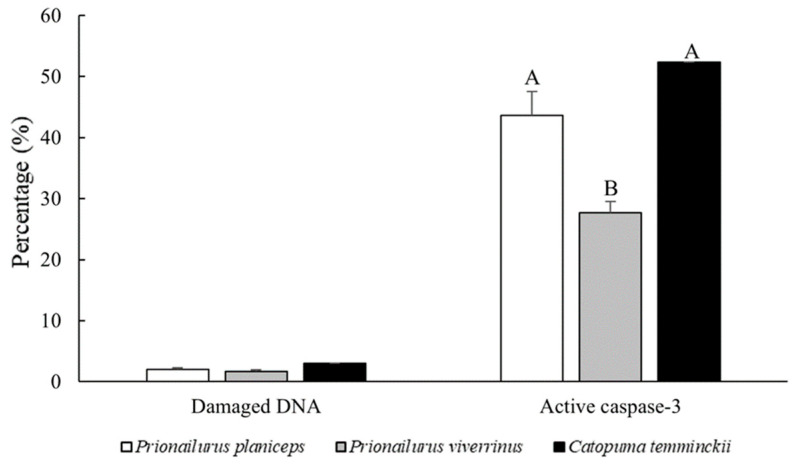
Comparisons of ejaculated semen traits among the flat-headed cat (*Prionailurus planiceps*), fishing cat (*Prionailurus viverrinus*), and Asiatic golden cat (*Catopuma temminckii*) in damaged DNA integrity and positive cleaved caspase-3. Bars with different superscripts within the same row are significantly different across species (*p* < 0.05).

**Figure 3 animals-14-01027-f003:**
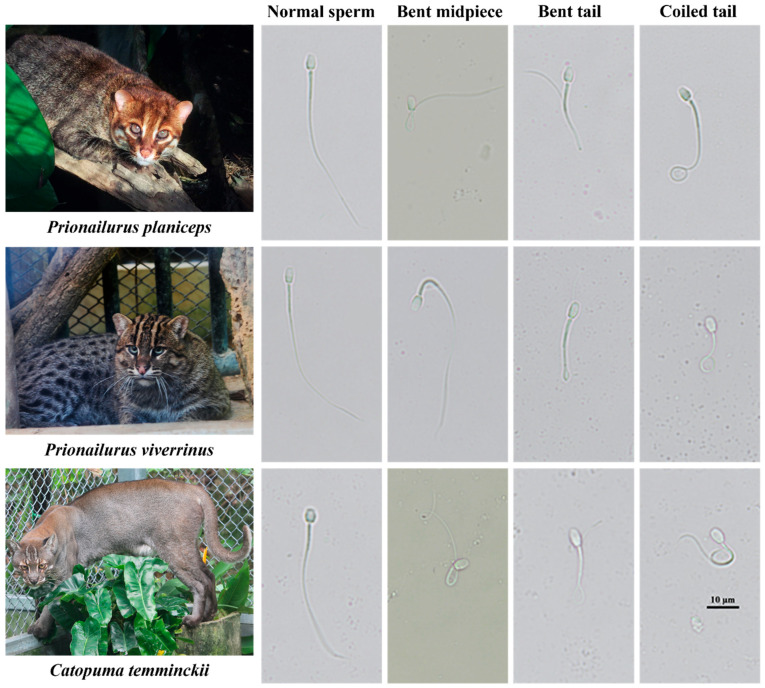
Photomicrographs of normal and frequently abnormal spermatozoa observed in the flat-headed cat (*Prionailurus planiceps*), fishing cat (*Prionailurus viverrinus*), and Asiatic golden cat (*Catopuma temminckii*) ejaculates (magnification: 1000×).

**Figure 4 animals-14-01027-f004:**
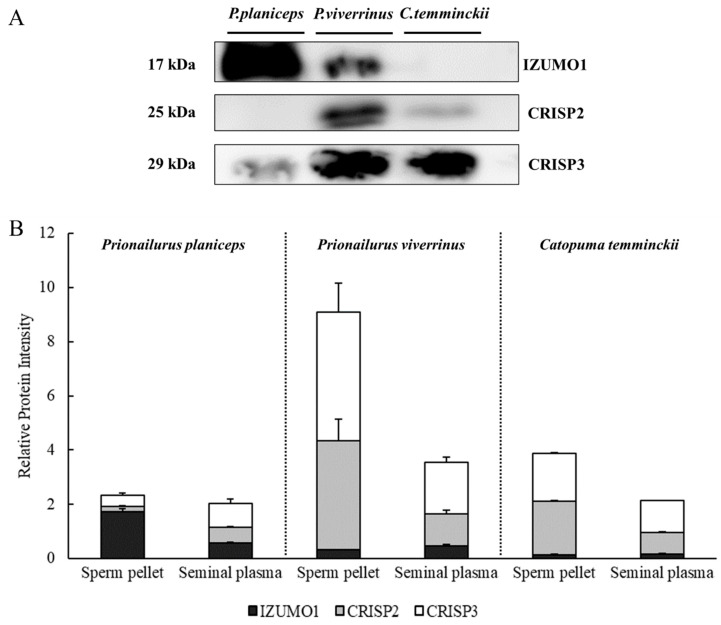
Representative Western immunoblotting for IZUMO1, CRISP2, and CRISP3 in sperm pellets among the flat-headed cat (*Prionailurus planiceps*) (*n* = 3), the fishing cat (*Prionailurus viverrinus*) (*n* = 3), and the Asiatic golden cat (*Catopuma temminckii*) (*n* = 1) (**A**). Graphs illustrating the comparison of their relative protein intensities (mean ± SEM) in both sperm pellets and seminal plasma in each species (**B**).

**Figure 5 animals-14-01027-f005:**
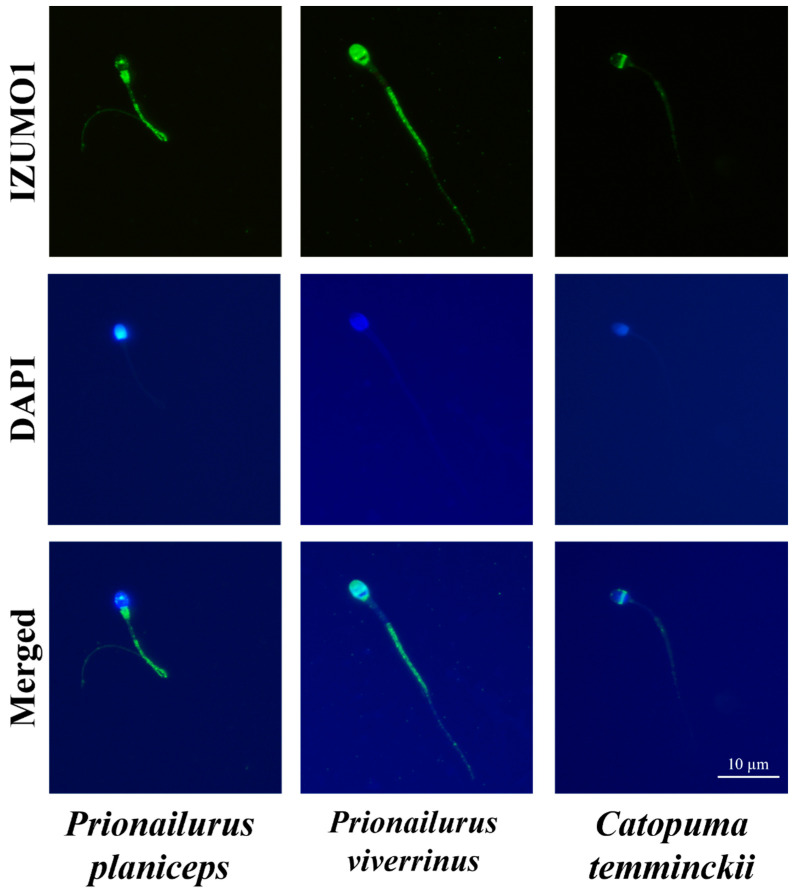
Immunofluorescence photomicrographs illustrate the localization of the IZUMO1 protein among the flat-headed cat (*Prionailurus planiceps*), the fishing cat (*Prionailurus viverrinus*), and the Asiatic golden cat (*Catopuma temminckii*) spermatozoa. The nuclear DNA was counterstained with DAPI (in blue), and the presence of IZUMO1 (in green) is visualized using a rabbit anti-IZUMO1 antibody and Alexa Fluor 488 dye-conjugated goat anti-rabbit IgG antibody (magnification: 1000×).

**Figure 6 animals-14-01027-f006:**
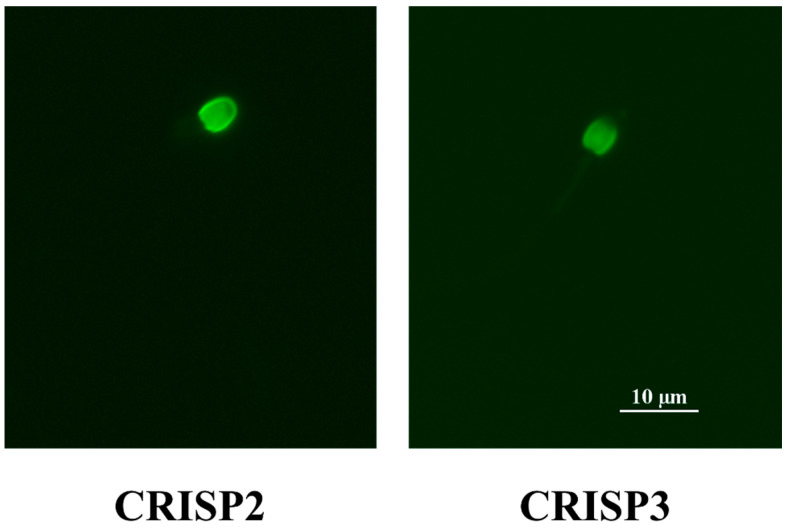
Immunofluorescence photomicrographs illustrate the presence of CRISP2 and CRISP3 in spermatozoa of the fishing cat (*Prionailurus viverrinus*). The presence of CRISP2 and CRISP3 (in green) is visualized using rabbit anti-CRISP2 and CRISP3 antibodies, respectively, and an Alexa Fluor 488 dye-conjugated goat anti-rabbit IgG antibody (magnification: 1000×).

**Figure 7 animals-14-01027-f007:**
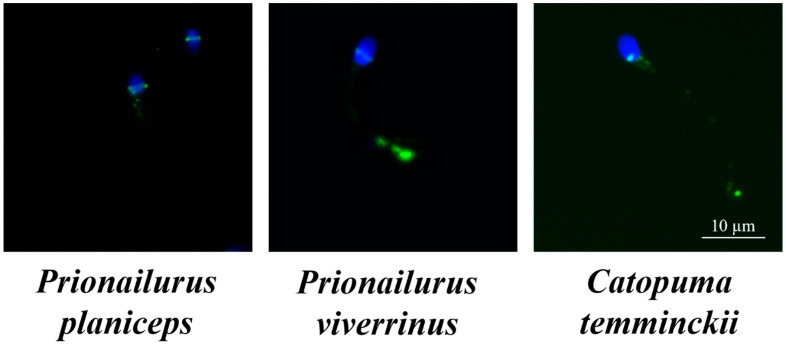
Immunofluorescence photomicrographs illustrate the presence of cleaved caspase-3 in spermatozoa of *Prionailurus planiceps*, *Prionailurus viverrinus*, and *Catopuma temminckii*. The nuclear DNA was counterstained with DAPI (in blue), and the presence of cleaved caspase-3 (in green) is visualized using a rabbit anti-cleaved caspase-3 antibody and Alexa Fluor 488 dye-conjugated goat anti-rabbit IgG antibody (magnification: 1000×).

**Table 1 animals-14-01027-t001:** Correction analysis results predicting the relationships between IZUMO1, CRISP2, and CRISP3 protein expression levels in sperm pellets and seminal plasma, and ejaculate characteristics of three wildcat species (*Prionailurus planiceps*, *Prionailurus viverrinus*, and *Catopuma temminckii*).

Parameters	IZUMO1		CRISP2		CRISP3	
	Sperm Pellet	Seminal Plasma	Sperm Pellet	Seminal Plasma	Sperm Pellet	Seminal Plasma
Semen volume (µL)	0.229 (*p* = 0.361)	−0.004 (*p* = 0.987)	0.253 (*p* = 0.309)	0.385 (*p* = 0.085)	0.536 (*p* < 0.05)	0.197 (*p* = 0.393)
Sperm concentration (spermatozoa/mL)	0.492 (*p* < 0.05)	0.495 (*p* < 0.05)	0.198 (*p* = 0.432)	0.275 (*p* = 0.227)	0.479 (*p* < 0.05)	0.137 (*p* = 0.563)
Total sperm number (spermatozoa)	0.492 (*p* < 0.05)	0.495 (*p* < 0.05)	0.198 (*p* = 0.432)	0.275 (*p* = 0.227)	0.479 (*p* < 0.05)	0.137 (*p* = 0.563)
Motility (%)	0.582 (*p* < 0.01)	0.596 (*p* < 0.01)	0.500 (*p* < 0.05)	0.367 (*p* = 0.102)	0.280 (*p* = 0.185)	0.175 (*p* = 0.447)
Viability (%)	0.441 (*p* < 0.05)	0.484 (*p* < 0.05)	0.615 (*p* < 0.01)	0.397 (*p* = 0.075)	0.402 (*p* = 0.052)	0.291 (*p* = 0.201)
Intact acrosome (%)	0.221 (*p* = 0.3)	0.739 (*p* < 0.01)	0.355 (*p* = 0.089)	0.503 (*p* < 0.05)	0.047 (*p* = 0.826)	0.137 (*p* = 0.563)
Normal sperm membrane (%)	0.566 (*p* < 0.01)	0.620 (*p* < 0.01)	−0.057 (*p* = 0.681)	0.004 (*p* = 0.975)	0.372 (*p* < 0.01)	0.037 (*p* = 0.775)
Normal sperm morphology (%)	0.524 (*p* < 0.05)	0.389 (*p* = 0.081)	−0.028 (*p* = 0.912)	0.279 (*p* = 0.220)	0.367 (*p* = 0.134)	0.090 (*p* = 0.697)
Normal DNA (%)	0.254 (*p* = 0.266)	0.548 (*p* < 0.05)	0.572 (*p* < 0.01)	0.548 (*p* < 0.05)	0.286 (*p* = 0.209)	0.153 (*p* = 0.508)

## Data Availability

Data supporting the findings of this study are available from the corresponding author upon reasonable request.

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
