# Peer review of "Detection of Protein Biomarkers Relevant to Sperm Characteristics and Fertility in Semen in Three Wild Felidae: The Flat-Headed Cat (Prionailurus planiceps), Fishing Cat (Prionailurus viverrinus), and Asiatic Golden Cat (Catopuma temminckii)"

_animals, 2024, doi:10.3390/ani14071027_

Round 1
Reviewer 1 Report
Comments and Suggestions for Authors
Overall, the study was well planned and carried through; the information given in the text provides a good background as well as perspectives of the detection of protein biomarkers related to the fertilization process, IZUMO1 and the CRISP-protein family, and the apoptotic. marker, active or cleaved caspase-3, in semen samples collected from these wild cats
Comments and Suggestions for Authors
Line 166: Add an r to canoy´s (Carnoy's)
Lines 286-287, 301: Add the protein expression levels of IZUMO1 CRISP2 and CRISP3 in domestic cats.
Lines 323, 325 Add percentage of positive cleaved caspase-3 in domestic cat spermatozoa.
¿Why were domestic cat semen samples not collected by electroejaculation?
¿Why were semen samples from domestic cats not collected by electroejaculation?
Figure 3, Figure 4: add common and scientific names.
Author Response
We thank the Reviewers for a detailed critique of our manuscript. We believe these comments have significantly improved the overall quality of our work. We hope we have addressed all the comments adequately to be acceptable for publication in Animals.
Responses to Specific Comments:
Reviewer #1:
Overall, the study was well planned and carried through; the information given in the text provides a good background as well as perspectives of the detection of protein biomarkers related to the fertilization process, IZUMO1 and the CRISP-protein family, and the apoptotic. marker, active or cleaved caspase-3, in semen samples collected from these wild cats
We thank this Reviewer for the favorable comments. We are delighted to note that the Reviewer found our manuscript well written and interesting.
Comments and Suggestions for Authors
Line 166: Add an r to canoy´s (Carnoy's)
Response: Per the Reviewer’s comment, we have made the change (page 4; line 167).
Lines 286-287, 301: Add the protein expression levels of IZUMO1 CRISP2 and CRISP3 in domestic cats.
Response: Per the Reviewer’s suggestions, we have included the protein expression levels of IZUMO1, CRISP2 and CRISP3 in domestic cats in the revised manuscript (please see Revised Manuscript page 8; line 292-294 and page 10; line 308-309).
Lines 323, 325 Add percentage of positive cleaved caspase-3 in domestic cat spermatozoa.
Response: Per the Reviewer’s suggestions, we have included the percentage of positive cleaved caspase-3 in domestic cat spermatozoa in the revised manuscript (please see Revised Manuscript page 11; line 338-340).
Why were domestic cat semen samples not collected by electroejaculation?
Why were semen samples from domestic cats not collected by electroejaculation?
Response: Since ethical approval for this study was specifically obtained for the three wild cat species and not for domestic cats, we decided to use epididymal sperm samples from routinely castrated domestic male cats for the protein expression comparison in this study. These samples were exempt from ethical approval requirements.
Figure 3, Figure 4: add common and scientific names.
Response: Per the Reviewer’s comment, we have added the common and scientific names in these figures (please see Revised Manuscript figures 4 and 5).

Reviewer 2 Report
Comments and Suggestions for Authors
This manuscript is well written, the subject is within the scope of the journal, the experimental design is over all adequate and the results provide interesting information regarding the occurrence of some protein biomarkers (IMZU, CRISP 2, CRISP 3 and cleaved-caspase 3) in the semen in three wild Felidae (Flat-headed cat, Fishing cat and Asiatic golden Cat). There are however, some minor correction and also aspects that need further clarification.
Materials and Methods
Line 132 – 134 “After semen collection, ejaculates with similar characteristics (sperm motility and viability) were pooled and kept in a Styrofoam container stored at ambient temperature until the evaluation of sperm quality was performed.”
How many collections per animal? How many ejaculates/animals? The pooled ejaculates were from different individuals?
Line 136-137 – It should be explained if it was only 1 domestic cat or more? What was the age of the cat/cats? Sperm was obtained only from the epididymal tail?
Line 166 – “…fixed in Canoy’s solution…” it should read “Carnoy’s”.
Lin2 176-178 – “Sperm pellets from domestic cats with good sperm quality (motility >80%) and low sperm quality (sperm motility <40%) were used for comparison”. It should be clarified if this sperm pellets samples are from the epididymal sperm described in line 136-137?
Results
Figure 1 – Separate A and B in two different figures, so that A and B is not repeated for different things in the same figure.
Line 259-264 – As these are wild species, it is inevitable that the number of samples available for the study are very low. However, the Asiatic golden cat only had 1 individual vs 3 individuals for the flat headed and fishing cat. What is the meaning of the statistical analysis between data from a species with only 1 individual (N=1) with the other species in which 3 individuals (N=3).
Figure 3 – Indicate the sample size/species.
Line 285-287 – Where is the data/results from the domestic cat sperm?
Line 303 - Add images for CRISP2 and CRISP 3 immunostaining.
Line 305 – 3.3 “Correlation between IZUMO1, CRISP2 …..”
Correlation analysis was done between parameters that were analysed before pooling of samples (motility and viability) and others after pooling of samples. Can authors explain, how this correlation analysis was not influenced by the pooling of ejaculates?
Discussion
Line 367-368 In my opinion this assumption is a bit of a stretch for the data of this study.
In the discussion section, it should be included the limitations of the study in particular regarding the sample size and the age of the animals, that could have influenced the results.
Author Response
We thank the Reviewers for a detailed critique of our manuscript. We believe these comments have significantly improved the overall quality of our work. We hope we have addressed all the comments adequately to be acceptable for publication in Animals.
Reviewer #2:
This manuscript is well written, the subject is within the scope of the journal, the experimental design is over all adequate and the results provide interesting information regarding the occurrence of some protein biomarkers (IMZU, CRISP 2, CRISP 3 and cleaved-caspase 3) in the semen in three wild Felidae (Flat-headed cat, Fishing cat and Asiatic golden Cat). There are, however, some minor correction and also aspects that need further clarification.
We thank this Reviewer for the favorable comments. We are delighted to note that the Reviewer found our manuscript well written and interesting.
Materials and Methods
Line 132 – 134 “After semen collection, ejaculates with similar characteristics (sperm motility and viability) were pooled and kept in a Styrofoam container stored at ambient temperature until the evaluation of sperm quality was performed.”
How many collections per animal? How many ejaculates/animals? The pooled ejaculates were from different individuals?
Response: Approximately 3 to 4 ejaculates per animal were collected, depending on the number of series of electroejaculation (EEJ). Only ejaculates from each individual animal that exhibited similar qualities (concentration, motility, and pH) were pooled together. We have also included this information in the revised manuscript (please see Revised Manuscript page 3; line 132).
Line 136-137 – It should be explained if it was only 1 domestic cat or more? What was the age of the cat/cats? Sperm was obtained only from the epididymal tail?
Response: The sperm samples from the cauda (tail) epididymis of six domestic cats were collected, comprising three cats with good sperm quality and three cats with poor sperm quality. Their ages range from approximately 1 to 4 years. We have also included this information in the revised manuscript (please see Revised Manuscript page 3; line 136-140).
Line 166 – “…fixed in Canoy’s solution…” it should read “Carnoy’s”.
Response: Per the Reviewer’s comment, we have made the change (page 4; line 167).
Lin2 176-178 – “Sperm pellets from domestic cats with good sperm quality (motility >80%) and low sperm quality (sperm motility <40%) were used for comparison”. It should be clarified if this sperm pellets samples are from the epididymal sperm described in line 136-137?
Response: According to the Reviewer’s concern, we have moved this sentence to the section that describes these samples earliest in the revised manuscript for clearer understanding (please see Revised Manuscript page 3; line 136-140).
Results
Figure 1 – Separate A and B in two different figures, so that A and B is not repeated for different things in the same figure.
Response: According to the Reviewer’s suggestions we have split figure 1A and 1B into figures 1 and 2 in the revised manuscript.
Line 259-264 – As these are wild species, it is inevitable that the number of samples available for the study are very low. However, the Asiatic golden cat only had 1 individual vs 3 individuals for the flat headed and fishing cat. What is the meaning of the statistical analysis between data from a species with only 1 individual (N=1) with the other species in which 3 individuals (N=3).
Response: We appreciate the Reviewer's concern regarding the small number of animals included in the experiment. To address this issue, we utilized different pooled samples derived from ejaculations showing varying semen quality from each individual animal, meaning that there were more than one sample per animal in this study, for statistical analyses. Additionally, for western blotting and immunofluorescent analysis, all samples were assessed in triplicate, providing a sufficient number of samples for statistical analyses.
Figure 3 – Indicate the sample size/species.
Response: According to the Reviewer’s suggestions, we have indicated the sample size/species information in the legend of this figure (Figure 4 in the revised manuscript).
Line 285-287 – Where is the data/results from the domestic cat sperm?
Response: Per the Reviewer’s question, we have included the protein expression levels of IZUMO1, CRISP2 and CRISP3 in domestic cats in the revised manuscript (please see Revised Manuscript page 8; line 292-294 and page 10; line 308-309).
Line 303 - Add images for CRISP2 and CRISP 3 immunostaining.
Response: Per the Reviewer suggestion, we have added the CRISP2 and CRISP3 immunostaining images in Figure 6 (revised version).
Line 305 – 3.3 “Correlation between IZUMO1, CRISP2 …..”
Correlation analysis was done between parameters that were analysed before pooling of samples (motility and viability) and others after pooling of samples. Can authors explain, how this correlation analysis was not influenced by the pooling of ejaculates?
Response: Correlation analysis was performed on parameters that were analyzed after pooling samples. We believed that this was not influenced by the semen pooling process, as only ejaculates from each individual animal with similar qualities (concentration, motility, and pH) were pooled together.
Discussion
Line 367-368 In my opinion this assumption is a bit of a stretch for the data of this study.
Response: We agree with the Reviewer's suggestions and have thus chosen to exclude this sentence from the revised manuscript.
In the discussion section, it should be included the limitations of the study in particular regarding the sample size and the age of the animals, that could have influenced the results.
Response: We appreciate the Reviewer's suggestions and have incorporated them into the discussion section of our revised manuscript (please see Revised Manuscript page 13; line 442-445).

Reviewer 3 Report
Comments and Suggestions for Authors
Very well written paper that is well described and flows nicely. A few comments below.
Line 137: how many domestic cat samples were used as a comparison?
Line 148/149: The EN stain does not really tell you if the sperm is dead or alive, only if the membrane is intact or not intact. Please change the "live" description to "membrane was intact at the time of staining" and "dead" to sperm membrane was not intact"
Line 175-176: where did they good sperm quality cats and poor quality sperm cats come from?
In all figures, list the common cat names since that is what you refer to in the text (rather than scientific names). It is easier for the reader to use the common names.
In materials and methods, list the known fertility of each of the cats used. Have they sired pregnancies? how many females were they bred to? When was the last litter sired?
Author Response
We thank the Reviewers for a detailed critique of our manuscript. We believe these comments have significantly improved the overall quality of our work. We hope we have addressed all the comments adequately to be acceptable for publication in Animals.
Reviewer #3:
Very well written paper that is well described and flows nicely. A few comments below.
We thank this Reviewer for the favorable comments. We are delighted to note that the Reviewer found our manuscript well written and interesting.
Line 137: how many domestic cat samples were used as a comparison?
Response: The sperm samples of from the cauda epididymis of six domestic cats were collected, comprising three cats with good sperm quality and three cats with poor sperm quality. We have also included this information in the revised manuscript (please see Revised Manuscript page 3; line 136-140).
Line 148/149: The EN stain does not really tell you if the sperm is dead or alive, only if the membrane is intact or not intact. Please change the "live" description to "membrane was intact at the time of staining" and "dead" to sperm membrane was not intact"
Response: Thank you for reviewer’s suggestion. We have made the recommended changes (Page 4, lines 149-151, Revised version).
Line 175-176: where did they good sperm quality cats and poor quality sperm cats come from?
Response: The domestic cat sperm samples were collected from the epididymis of six castrated males provided by the Veterinary Clinic. After collection, they were evaluated for sperm quality and divided into two categories: good quality cat sperm (motility ≥ 80%) and low quality cat sperm (motility < 40%). This detail has been included in the revised version (please see Revised Manuscript page 3; line 136-140).
In all figures, list the common cat names since that is what you refer to in the text (rather than scientific names). It is easier for the reader to use the common names.
Response: Thank you for reviewer’s suggestion, we have added the common and scientific names in all figures (please see Revised Manuscript).
In materials and methods, list the known fertility of each of the cats used. Have they sired pregnancies? how many females were they bred to? When was the last litter sired?
Response: In the history of fertility, while all three male fishing cats were paired with each female, only one male fishing cat successfully produced one offspring in early 2024. Similarly, all the male flat-headed cats were paired with a single aging female (> 10 years old), resulting in the successful birth of one offspring in 2018. The male Asiatic golden cat has never been paired with females due to the unavailability of any female cats. Given these limitations in the number and age of females, we believe that the history of fertility does not provide a comprehensive reflection of their actual fertility. Therefore, we have chosen not to include this information in the materials and methods section.
